# Preference-based Multi-Objective Bayesian Optimization with Gradients

**Joshua Hang Sai Ip**[1]     **Ankush Chakrabarty**[2]     **Hideyuki Masui**[3]

**Ali Mesbah**[1]     **Diego Romeres**[2]

[1]University of California, Berkeley    [2]Mitsubishi Electric Research Laboratories
[3]Mitsubishi Electric

[1]{ipjoshua, mesbah}@berkeley.edu [2]{chakrabarty, romeres}@merl.com
[3]masui.hideyuki@bc.mitsubishielectric.co.jp

## Abstract

We propose PUB-MOBO for personalized multi-objective Bayesian Optimization. PUB-MOBO combines utility-based MOBO with local multi-gradient descent to refine user-preferred solutions to be near-Pareto-optimal. Unlike traditional methods, PUB-MOBO does not require estimating the entire Pareto-front, making it more efficient. Experimental results on synthetic and real-world benchmarks show that PUB-MOBO consistently outperforms existing methods in terms of proximity to the Pareto-front and utility regret.

## 1 Introduction

Multi-objective Bayesian optimization (MOBO) is a particularly useful multi-objective optimization (MOO) strategy when the objectives are black-box functions constructed from noisy observations. Traditional MOBO methods such as $q$-EHVI [1] assume that all Pareto-optimal solutions are equally desirable to the user, which might not be the case in practice. There has been a growing interest in preference-based MOBO (e.g., [2, 3]) that leverages user preferences to guide the optimization process towards regions of interest within the Pareto-front, typically in the form of pairwise comparisons between solutions generated by the optimization algorithm. These comparisons are used to estimate an underlying utility function that describes user preferences. In [4], the authors propose the EUBO and qEIUU acquisition functions respectively, which take advantage of user-preference when querying new points. However, while preference-based MOBO can effectively identify solutions with high utility as informed by user feedback, the resulting solutions may not be Pareto-optimal.

We present Preference-Utility-Balanced MOBO (PUB-MOBO), that systematically determines the user-informed regions of interest within the Pareto-front by synergizing global and local search strategies. PUB-MOBO begins with a global search driven by utility maximization to identify regions in the solution space that align with user preferences. Subsequently, a local search is conducted in the vicinity of these solutions to discover dominating solutions that are closer to Pareto-optimality. Additionally, a new utility function, the Preference-Dominated Utility Function (PDUF), is proposed that encapsulates the concept of dominance within a single function. PDUF allows for consistently identifying dominating solutions, while providing a straightforward means for expressing all possible user preferences. This differs from existing utility functions for preference-based MOBO such as the negative $\ell_1$ distance from an ideal solution irrespective of the solution being on the Pareto-front or an infeasible ideal solution [5], or the weighted sum where not all Pareto-optimal points can be assigned with the highest utility value from any choice of weights [6]. PDUF is then used in conjunction with gradient descent (GD) to seamlessly combine user preferences with the notion of dominance to identify user-preferred solutions that are approximately Pareto-optimal. Empirical demonstrations on

---

several synthetic benchmark and real-world problems show that PUB-MOBO not only enhances the utility of the optimization solutions, but also yields near Pareto-optimal solutions.

## 2   Problem Formulation

We *minimize* $n_f$ expensive-to-evaluate objective functions, denoted by $f_i(\boldsymbol{x})$ for $i \in \{1, \cdots, n_f\}$. Consequently, the objective function vector is denoted $\boldsymbol{f}(\boldsymbol{x})$, where $\boldsymbol{x} \in \mathbb{R}^{n_x}$ denote the decision variables. We assume that for a candidate $\boldsymbol{x}$, the function $\boldsymbol{f}(\boldsymbol{x})$ can be evaluated, but no first- or higher-order information about any component of $\boldsymbol{f}$ is available. No analytical form of $\boldsymbol{f}$ is known.

For MOO problems without user-preferences, the objective is to attain Pareto-optimality, which is defined as follows [7]. Note that accurately computing the set of Pareto-optimal points, referred to as the Pareto-front $\mathbb{X}_{\text{pareto}}$, can often be computationally prohibitive, even for small $n_f$. In the presence of a user, estimating the entire Pareto-front may become unnecessary, especially when only specific sub-regions of the feasible set $\mathbb{X}$ is of interest. Mathematically, such user-preferences are often abstracted in the MOBO literature via *utility functions*. Specifically, the MOO problem is recast as a (scalar) utility maximization problem

$$\max_{\boldsymbol{x} \in \mathbb{X}} \quad u\left(\boldsymbol{f}(\boldsymbol{x})\right), \tag{1}$$

where $u : \mathbb{R}^{n_y} \to \mathbb{R}$ is the unknown utility function that dictates the behavior of the user. Note that the input to the utility is a noise-corrupted outcome vector $\boldsymbol{y} = \boldsymbol{f}(\boldsymbol{x}) + \varepsilon$, where $\varepsilon$ is zero-mean noise with variance $\sigma_\varepsilon^2 \boldsymbol{I}_{n_y}$ where $\boldsymbol{I}_{n_y}$ is the $n_y \times n_y$ identity matrix. Let the highest utility Pareto-point be defined as

$$\boldsymbol{x}^* \in \arg\max_{\boldsymbol{x} \in \mathbb{X}_{\text{pareto}}} u(\boldsymbol{f}(\boldsymbol{x})). \tag{2}$$

Following the preference BO literature, we assume the utility function is not available to evaluate and its functional form is unknown. Additionally, it is well-established that user preferences are difficult to be assigned to continuous numerical values; instead we suppose that users are more inclined to provide weak supervision in the form of pairwise comparisons [8, 4]. The following assumption is made to assert that a typical user will select dominating solutions when possible: *If $\boldsymbol{y}_1$ and $\boldsymbol{y}_2$ are candidate outcomes presented to the user and $\boldsymbol{y}_1 \succ \boldsymbol{y}_2$, then the user will always select $\boldsymbol{y}_1$; that is, $u(\boldsymbol{y}_1) > u(\boldsymbol{y}_2)$.* This assumption should be enforced when modeling preference-based MOBO problems to accurately reflect real user behavior.

## 3   Preference-Utility-Balanced (PUB) MOBO

Users often require some assurance that the suggested candidates are not only high in utility, but also near-(Pareto)-optimal. PUB-MOBO relies on utility maximization to ascertain candidate solutions that are preferred by the user while promoting a local search towards the Pareto-front using estimated gradients. We observe that the local search finds solutions near Pareto points, which subsequently accelerates the search for high-utility solutions.

### 3.1   PUB-MOBO Algorithm

The proposed PUB-MOBO method operates in three stages. We extend the two stages (PE: preference exploration, and EXP: outcome evaluation via experiments) in [4] with an additional stage based on local multi-gradient descent, denominated the GD stage. In each PUB-MOBO iteration, these three stages are executed, and the process is repeated *ad infinitum*, or (more practically) until a pre-decided budget for total number of outcome evaluations is attained; see Algorithm 1 in Appendix C.
**Preference Exploration**: Here, the user expresses their preferences over a query of two candidate solutions in a form of pairwise comparisons. The comparison is used to update the estimate $\hat{u}$ of the utility, obtained implicitly with a pairwise GP and the EUBO acquisition function proposed in [4]; see Appendix B.1 for the closed-form expression. Note that no evaluation of $\boldsymbol{f}$ is required for PE.
**Outcome evaluation via Experiments**: Here, we compute the optimal decision variables and evaluate true outcomes to update the outcome model $\hat{\boldsymbol{f}}$ using the expected improvement under utility uncertainty (qEIUU) [9] acquisition; see Appendix B.2 for the closed-form expression. Maximizing qEIUU involves taking Monte Carlo samples [10, 11] and yields the optimal decision variables, $\boldsymbol{x}_{\text{EXP}}$.

After $x_{\text{EXP}}$ is obtained, we append it along with its true outcome value $f(x_{\text{EXP}})$ to the current dataset.
**Multi-gradient descent**: This GD stage is motivated by the fact that $x_{\text{EXP}}$, while expected to be high in utility, is not specifically designed to be near the Pareto-front. Analogous to single-objective optimization, we will pursue local gradients that are expected to generate a trajectory of $x$ candidates that evolves towards a nearby Pareto-optimal point. We will refer to these gradient-following decision variables as '$x_{\text{GD}}$'. We set the initial $x_{\text{GD}}$ to be $x_{\text{EXP}}$.

For a MOO problem, gradient descent must be adapted for multiple objectives. We propose the use of multiple gradient descent algorithm (MGDA) [12], which was designed for smooth multi-outcome objective functions. MGDA exhibits some theoretical properties that, we hypothesize, and demonstrate via experiments, are beneficial in the MOBO context.

MGDA exploits the KKT conditions [13] as a quadratic cost constrained on the probability simplex:

$$\min_{\alpha \geq 0} \left\| \alpha^\top \nabla f(x) \right\|^2 \text{ subject to: } \mathbf{1}^\top \alpha = 1. \tag{3}$$

It is well-known, c.f. [12], that a solution to (3) is either: $\alpha^\top \nabla f(x) = 0$, in which case the current parameters $x$ are Pareto-optimal, or $\alpha^\top \nabla f(x) \neq 0$, and $\alpha^\top \nabla f(x)$ is a feasible descent direction. Given that (3) is a quadratic cost over linear constraints, we can use the Frank-Wolfe algorithm [14, 15] to efficiently compute optimal solutions; see pseudocode in Algorithm 3 in Appendix C.

Solving (3) yields an optimal $\alpha$ with which we can take a gradient step $x_{\text{GD}} \leftarrow x_{\text{GD}} - \eta \alpha^\top \nabla f(x_{\text{GD}})$. However, there are two clear difficulties at this juncture. The first is that this update may yield an $x_{\text{GD}} \notin \mathbb{X}$. To counter this, we stop updating when this happens, and stop the local gradient search phase, moving on to the next PUB-MOBO iterations with an updated dataset $D$ that contains all the $x_{\text{GD}}$ and correspond $y_{\text{GD}}$ observed so far. The second and more debilitating problem is that we do not have access to gradients of $f$. Thankfully, we do have a surrogate model $\hat{f}$ with which we can obtain an estimate of the gradient at any $x$ with $\mu^\nabla := \mathbb{E}[\nabla \hat{f}(x)]$ through (7a). The gradient step is then $x_{\text{GD}} \leftarrow x_{\text{GD}} - \eta \alpha^\top \mu^\nabla(x_{\text{GD}})$. Unfortunately, there is no clear correlation between the uncertainties in $f$ and $\nabla f$, so $\mu^\nabla$ could have large uncertainties even near previously observed points. Therefore, it is imperative to incorporate techniques that can reduce uncertainty in the posterior of the gradient estimate. To this end, we propose to use the gradient information (GI) acquisition function [16].

**Multi-gradient descent with GI acquisition**: We briefly explain the mechanism of the GI acquisition. Suppose we select the best candidate from the EXP stage, $x_{\text{EXP}}$, and set it as the initial candidate for the local gradient search: $x_{\text{GD}}$. The GI acquisition tries to select a subsequent point $x'$ that will minimize the uncertainty of the gradient at $x_{\text{GD}}$ if $x'$ and its corresponding $y'$ were known. By considering all $n_f$ objective independently distributed, we assess the uncertainty can formulate the uncertainty information using an A-optimal design criterion [17], which, for Gaussian distributions, involves maximizing:

$$\text{GI}(x') = \sum_{i=1}^{n_f} \text{Tr}\left( \nabla k_i(x_{\text{GD}}, X') \mathcal{K}_\sigma^{-1}(X') \nabla k_i^\top(x_{\text{GD}}, X') \right) \tag{4}$$

where $X' = \{X \cup x'\}$. For each gradient-step in $n_{\text{GD}}$, the GI acquisition function is optimized $n_{\text{GI}}$ times to reduce gradient uncertainty. Upon each optimization, we evaluate the outcome function to obtain a corresponding $y_{\text{GD}}$, which is appended to the dataset $D$ for subsequent PUB-MOBO iterations. We provide the derivation of the GI acquisition function in Appendix B.3 and pseudocode of multi-gradient descent in Algorithm 2, in Appendix C.

### 3.2 Preference-Dominated Utility Function

We propose the PDUF, which merges the concept of dominance with user preferences to help locate high utility points that are close to Pareto-optimality. The utility function, which represents user preferences, is employed to respond to user queries, such as providing pairwise comparisons between two outcomes [4]. It should satisfy two key properties:

(P1) *Dominance Preservation*: When evaluating a query, the true utility function should satisfy Assumption 1.

(P2) *Preference Integration*: The utility function should have parameters $\theta_u$ that allow unique strictly maximal-utility Pareto-optimal solutions. That is, for any $x \in \mathbb{X}_{\text{pareto}}$, there exists an easily computable $\theta_u \in \mathbb{R}^{n_u}$ such that $u(f(x)|\theta_u) > u(f(\{\mathbb{X}_{\text{pareto}} \setminus x\}|\theta_u)$.

For instance, the commonly used $\ell_1$ distance (a) fails to satisfy the *Preference Integration* property when calculated from the utopia point, and violates *Dominance Preservation* when calculated from any other point. This is illustrated in Fig. 1a, where the contours of an $\ell_1$ distance utility function is shown with an example Pareto-front. Here, the two red points are indistinguishable according to the utility function, demonstrating the limitations of $\ell_1$ distance in distinguishing between Pareto-optimal solutions.

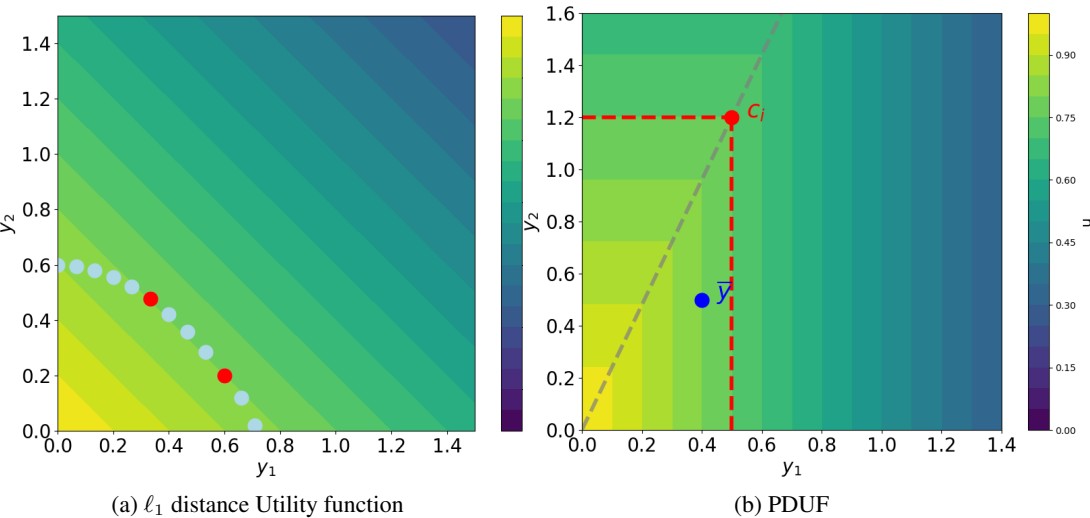

(a) $\ell_1$ distance Utility function           (b) PDUF

Figure 1: Contour plots of (a) the commonly used negative $l_1$ distance Utility function (b) the proposed PDUF.

Therefore, we propose the preference-dominated utility function (PDUF) which merges the concept of dominance with user preferences. An illustration of the contours in a 2D case is shown in Fig. 1b. The PDUF integrates the concept of dominance with user preferences by combining multiple logistic functions centered around different points in the objective function space and is expressed as:

$$u(\boldsymbol{y}) = \frac{1}{n_c} \sum_{i=1}^{n_c} \prod_{j=1}^{n_y} L_\beta(y_j, c_{i,j}) \tag{5}$$

where $L_\beta(y_j, c_{i,j}) = \frac{1}{1+\exp(\beta \cdot (y_j - c_{i,j}))}$. $c_i = (c_{i,1}, c_{i,2}, \dots, c_{i,n_y})$ denotes the $i^{\text{th}}$ center for one logistic function, $\beta$ denotes a parameter that controls the steepness of the logistic function, and $n_c$ denotes the number of centers. The logistic function $L_\beta(y_j, c_{i,j})$ approximates the step function and enforces dominance for each objective $y_j$, as seen in the red dashed lines in Fig. 1b, and the product aggregates this approximation for all objectives. Furthermore, the sum of logistic function products preserve dominance in the objective space. Indeed, for every $\bar{\boldsymbol{y}}$ that dominates user query $\boldsymbol{c}_i$, PDUF will express user preference with $u(\bar{\boldsymbol{y}}) > u(\boldsymbol{c}_i)$. Finally, the centers define the parameters $\theta_u$ that ensure the utility function adheres to the *preference integration* property by aligning them along an arbitrary line (the grey line in Fig. 1b).

## 4   Experiments

We validate the proposed PUB-MOBO method on benchmarks commonly found in MOO literature: DTLZ1 ($n_x = 9, n_f = 2$) [18], DH1 ($n_x = 10, n_f = 2$) [7], Conceptual Marine Design ($n_x = 6, n_f = 4$) [19], Car Side Impact ($n_x = 7, n_f = 4$) [20]. The baselines and ablations that we compare are (i) EUBO+qEIUU baseline which contains only the PE and EXP stages; (ii) PUB-MOBO-PG which uses the predicted gradients (PG) without any outcome evaluations or GI optimizations in the GD stage. This makes it relatively inexpensive, but it ignores the fact that additional samples can yield useful derivative information; (iii) PUB-MOBO-PG+OE which is a PUB-MOBO ablation that uses the predicted gradients as in PUB-MOBO-PG, but an Outcome Evaluation (OE) is performed at every gradient descent step in an effort to lower gradient uncertainty around observed points; (iv)

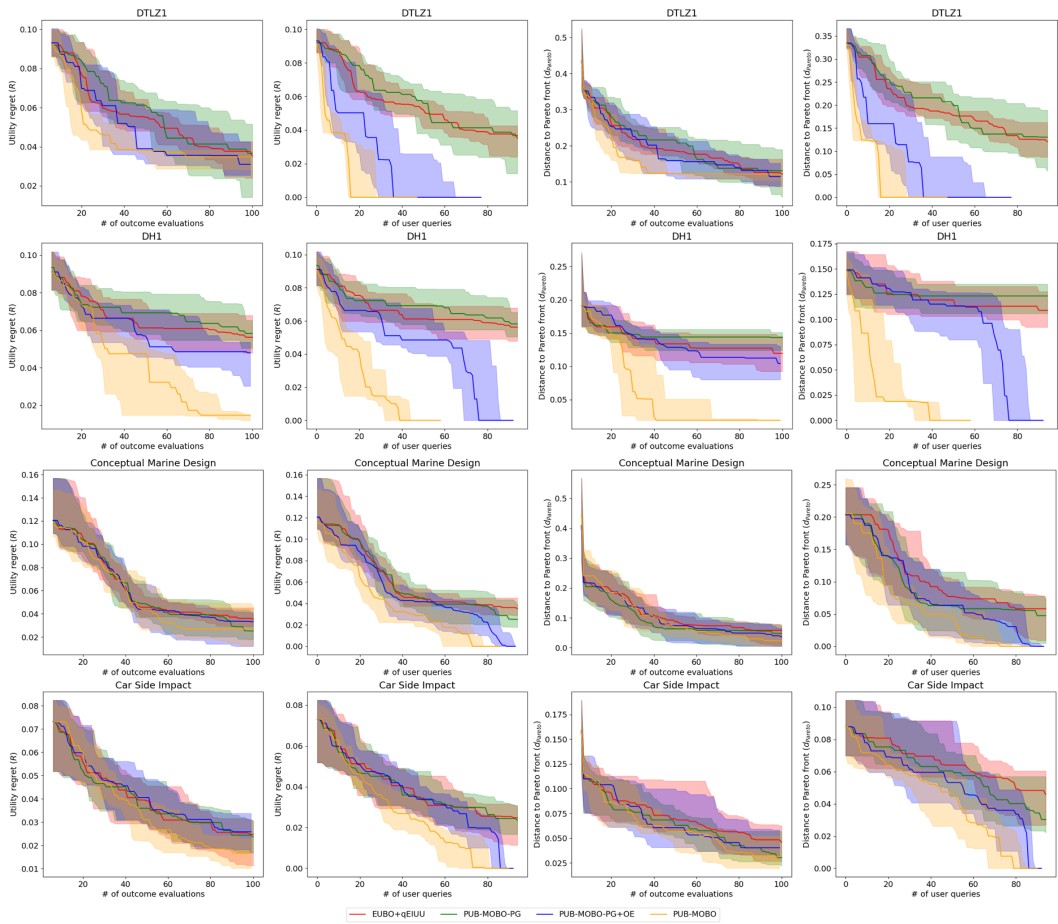

Figure 2: Performance comparison on benchmarks DTLZ1, DH1, Conceptual Marine Design, Car Side Impact. Continuous lines show median over 100 runs, and shading indicates 25-75 percentiles.

PUB-MOBO which is the proposed method. Figure 2 illustrates the performance of the experiments in terms of utility regret and distance to the Pareto front w.r.t. outcome evaluations and user queries.

EUBO+qEIUU is the poorest-performing algorithm for all the metrics affirming the effectiveness of the additional stage based on local gradient search. However, PUB-MOBO-PG performs equally poorly, largely due to inaccurate gradient estimation obtained with the surrogate model $\hat{\boldsymbol{f}}$ in (7). We frequently observe that the evolution of $\boldsymbol{x}_{\mathrm{GD}}$ in the GD stage is prematurely terminated either due to infeasibility in $\boldsymbol{x}$ or because of incorrect solutions to MGDA due to erroneous $\boldsymbol{\mu}^{\nabla}(\boldsymbol{x}_{\mathrm{GD}})$. The PG+OE variant significantly outperforms the PG variant due to its enhanced gradient estimation accuracy, which justifies the additional computational cost of updating the outcome model. PUB-MOBO further improves on the PG+OE variant by using the GI acquisition function to reduce gradient uncertainty, leading to even more accurate gradient estimates.

## 5   Conclusion

In this work we presented PUB-MOBO, a sample efficient Multi-Objective Bayesian Optimization algorithm that combine user-preference with a gradient-based search to compute near Pareto-optimal solutions. We verify that our proposed method yields high utility and reduced distance to Pareto-front solutions, and also demonstrate the importance of gradient uncertainty reduction in the gradient-based search. Finally, the proposed utility function respects dominance while modeling different user preferences.

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

# Appendix

## A    Preliminaries

### A.1    Modeling with Gaussian processes

We first discuss the modeling choices considered to learn the outcome function $\boldsymbol{f}$ and the utility function $u$: their respective approximations are denoted $\hat{\boldsymbol{f}}$ and $\hat{u}$.

#### A.1.1    Modeling outcomes

Gaussian process (GP) regression is a popular choice for constructing the surrogate $\hat{\boldsymbol{f}}$ for the true outcome function $\boldsymbol{f}$. We train an independent GP for each objective $\boldsymbol{f}_i$, though a multi-output GP that models correlations between the objectives could also be considered [21]. Each GP is defined *a priori* by a mean function $m(\boldsymbol{x})$ and covariance function $k_i(x, x')$ called kernel. For this work, any $\mathcal{C}^2$ kernel is admissible.

Let $\boldsymbol{X}_T = [\boldsymbol{x}_1, \boldsymbol{x}_2, \ldots, \boldsymbol{x}_T]$; we drop the subscript for brevity. Given a dataset $D := (\boldsymbol{X}, \boldsymbol{Y})$, comprising input-outcome pairs, the mean and variance of the posterior are given by

$$\mu_i(\boldsymbol{x}) = m(\boldsymbol{x}) + k_i(\boldsymbol{x}, \boldsymbol{X})\mathcal{K}_\sigma^{-1}(\boldsymbol{X})(Y_i - m(\boldsymbol{X})), \tag{6a}$$

$$\Sigma_i(\boldsymbol{x}) = k_i(\boldsymbol{x}, \boldsymbol{x}) - k_i(\boldsymbol{x}, \boldsymbol{X})\mathcal{K}_\sigma^{-1}(\boldsymbol{X})k_i(\boldsymbol{X}, \boldsymbol{x}), \tag{6b}$$

where $\mathcal{K}_\sigma(\boldsymbol{X}) := k_i(\boldsymbol{X}, \boldsymbol{X}) + \sigma^2 \boldsymbol{I}$ and $m(\cdot)$ is the prior mean. Since the derivative is a linear operator, the derivative GP is another GP [22] characterized fully by the mean and covariance functions

$$\mu_i^\nabla(\boldsymbol{x}) = \nabla m(\boldsymbol{x}) + \nabla k_i(\boldsymbol{x}, \boldsymbol{X})\mathcal{K}_\sigma^{-1}(\boldsymbol{X})(Y_i - m(\boldsymbol{X})), \tag{7a}$$

$$\Sigma_i^\nabla(\boldsymbol{x}) = \nabla^2 k_i(\boldsymbol{x}, \boldsymbol{x}) - \nabla k_i(\boldsymbol{x}, \boldsymbol{X})\mathcal{K}_\sigma^{-1}(\boldsymbol{X})\nabla k_i(\boldsymbol{X}, \boldsymbol{x}), \tag{7b}$$

In the implementation, each GP is designed with a Matérn 5/2 kernel with ARD, a lengthscale prior defined by Gamma($\alpha = 3$, $\beta = 6$), and an outputscale prior defined by Gamma($\alpha = 2$, $\beta = 0.15$). To infer the gradient mean (7a) and covariance (7b) of the posterior, automatic differentiation [23] is used. The inputs $\boldsymbol{X}$ are normalized from [0, 1] and the outcomes $\boldsymbol{Y}$ are standardized to zero mean and unit variance during GP fitting. We intialize the model with 6 outcomes.

#### A.1.2    Modeling preferences

We assume the user is only capable of weak supervisions in the form of pairwise comparisons (PC). That is, if the user prefers $\boldsymbol{y} := \boldsymbol{f}$ over $\boldsymbol{y} := \boldsymbol{f}'$, the pairwise comparison function $r(\boldsymbol{y}, \boldsymbol{y}') = 0$. In the event that the user prefers $\boldsymbol{y}'$ instead, $r(\boldsymbol{y}, \boldsymbol{y}') = 1$. Pairwise GPs c.f. [8] allow us to learn a latent functional representation $\hat{u}$ of the true user utility based on this preference feedback. The latent function satisfies $\hat{u}(\boldsymbol{y}) > \hat{u}(\boldsymbol{y}')$ if the user prefers $\boldsymbol{y}$, and vice versa. In the implementation, we use the RBF kernel with ARD, a lengthscale prior defined by Gamma($\alpha = 2.4$, $\beta = 2.7$), and an outputscale prior defined by a smoothed box prior from [0.01, 100]. The outcomes $\boldsymbol{Y}$ are normalized from [0, 1] during GP fitting. We initialize with 12 Sobol points and form pairwise comparisons with every consecutive pair of outcomes to yield 6 user comparisons.

## B    Acquisition functions

### B.1    EUBO

The EUBO acquisition is given by:

$$\mathsf{EUBO}(\boldsymbol{x_1}, \boldsymbol{x_2}) = \mathbb{E}[\max(\hat{u}(\hat{\boldsymbol{f}}(\boldsymbol{x_1})), \hat{u}(\hat{\boldsymbol{f}}(\boldsymbol{x_2})))], \tag{8}$$

where the hat notation denotes surrogate models of the corresponding functions.

## B.2 qEIUU

The expected improvement under utility uncertainty is given by

$$\mathsf{qEIUU}(\boldsymbol{x}) = \mathbb{E}\left[\max(\hat{u}(\hat{\boldsymbol{f}}(\boldsymbol{x})) - \hat{u}(\boldsymbol{f}(\boldsymbol{x}_{\text{best}})), 0)\right], \tag{9}$$

where $\boldsymbol{x}_{\text{best}} = \arg\max_{x \in \boldsymbol{X}} \hat{u}(\hat{\boldsymbol{f}}(\boldsymbol{x}))$, and $\boldsymbol{x}_{\text{EXP}} := \arg\max_{\mathbb{X}} \mathsf{qEIUU}(\boldsymbol{x})$. Since the expectation in (9) is with respect to the outcome and utility models, the analytical expression is challenging.

## B.3 GI

We derive the GI acquisition function described in (4). The derivation is an adaptation of the Gradient Information acquisition function in [16] to the case of independent multi-objectives. We begin by expressing the difference in the trace of the gradient posterior covariance before and after the addition of the new datapoint $(\boldsymbol{x}', \boldsymbol{y}')$ to the dataset $D$

$$\text{GI} = \sum_{i=1}^{n_f} \mathbb{E}\left[\mathsf{Tr}(\Sigma_i^\nabla(\boldsymbol{x}_{\text{GD}}|D)) - \mathsf{Tr}\left(\Sigma_i^\nabla(\boldsymbol{x}_{\text{GD}}|D, (\boldsymbol{x}', \boldsymbol{y}')))\right)\right].$$

This can be expressed as the Lebesgue-Stieltjes integral

$$\text{GI} = \sum_{i=1}^{n_f} \int \left[\mathsf{Tr}(\Sigma_i^\nabla(\boldsymbol{x}_{\text{GD}}|D)) - \mathsf{Tr}\left(\Sigma_i^\nabla(\boldsymbol{x}_{\text{GD}}|D, (\boldsymbol{x}', \boldsymbol{y}')))\right)\right] dF(\boldsymbol{x}'),$$

with $F$ denoting the distribution of $\boldsymbol{x}'$. For optimization purposes maximizing GI is equivalent to maximizing

$$\underset{\boldsymbol{x}'}{\arg\max}\ \text{GI} \equiv \underset{\boldsymbol{x}'}{\arg\max} \sum_{i=1}^{n_f} \int -\mathsf{Tr}\left(\Sigma_i^\nabla(\boldsymbol{x}_{\text{GD}}|D, (\boldsymbol{x}', \boldsymbol{y}')))\right) dF(\boldsymbol{x}').$$

since the first term does not depend on the optimization variable $\boldsymbol{x}'$. Rewriting this formulation as a Reinmann integral will yield

$$\underset{\boldsymbol{x}'}{\arg\max}\ \text{GI} = \underset{\boldsymbol{x}'}{\arg\min} \sum_{i=1}^{n_f} \int_{\mathbb{R}} \mathsf{Tr}\left(\Sigma_i^\nabla(\boldsymbol{x}_{\text{GD}}|D, (\boldsymbol{x}', \boldsymbol{y}')))\right) \cdot p(\boldsymbol{f}(\boldsymbol{x}') = \boldsymbol{y}'|D) d\boldsymbol{y}'.$$

As seen from (7b), the covariance in Gaussian distributions is independent of the observed outcomes, so the acquisition function can be further reduced to

$$\underset{\boldsymbol{x}'}{\arg\max}\ \text{GI} = \underset{\boldsymbol{x}'}{\arg\min} \sum_{i=1}^{n_f} \mathsf{Tr}\left(\Sigma_i^\nabla(\boldsymbol{x}_{\text{GD}}|D, (\boldsymbol{x}', \boldsymbol{y}')))\right) \underbrace{\int_{\mathbb{R}} p(\boldsymbol{f}(\boldsymbol{x}') = \boldsymbol{y}'|D) d\boldsymbol{y}'}_{=1},$$

$$= \underset{\boldsymbol{x}'}{\arg\min} \sum_{i=1}^{n_f} \mathsf{Tr}\left(\Sigma_i^\nabla(\boldsymbol{x}_{\text{GD}}|D, (\boldsymbol{x}', \boldsymbol{y}')))\right),$$

$$= \underset{\boldsymbol{x}'}{\arg\max} \sum_{i=1}^{n_f} \mathsf{Tr}\left(\nabla k_i(\boldsymbol{x}_{\text{GD}}, \boldsymbol{X}')\mathcal{K}_\sigma^{-1}(\boldsymbol{X}')\nabla k_i^\top(\boldsymbol{x}_{\text{GD}}, \boldsymbol{X}')\right),$$

where $\boldsymbol{X}' = \{\boldsymbol{X} \cup \boldsymbol{x}'\}$.

## C  PUB-MOBO Algorithms

---

**Algorithm 1** PUB-MOBO

---

1: Generate initial data: $\boldsymbol{x}_{\text{INIT}}, \boldsymbol{y}_{\text{INIT}}, r(\boldsymbol{y}_{\text{INIT}})$
2: $D = (\boldsymbol{x}_{\text{INIT}}, \boldsymbol{y}_{\text{INIT}})$
3: $P = (\boldsymbol{y}_{\text{INIT}}, r(\boldsymbol{y}_{\text{INIT}}))$
4: Update outcome model $\hat{\boldsymbol{f}}$ with $(\boldsymbol{x}_{\text{INIT}}, \boldsymbol{y}_{\text{INIT}})$
5: Update preference model $\hat{u}$ with $(\boldsymbol{y}_{\text{INIT}}, r(\boldsymbol{y}_{\text{INIT}}))$
6: **while** # outcome evaluations $\leq$ budget **do**
7:     **PE stage**
8:     $\boldsymbol{x_1}, \boldsymbol{x_2} \leftarrow \text{argmax}_{\boldsymbol{x_1}, \boldsymbol{x_2}} \text{ EUBO}$
9:     $\boldsymbol{y_1}, \boldsymbol{y_2} = \hat{\boldsymbol{f}}(\boldsymbol{x_1}), \hat{\boldsymbol{f}}(\boldsymbol{x_2})$
10:    $r(\boldsymbol{y_1}, \boldsymbol{y_2}) \leftarrow$ user provides a comparison
11:    Append $P$ with $(\boldsymbol{y_1}, \boldsymbol{y_2}, r(\boldsymbol{y_1}, \boldsymbol{y_2}))$
12:    Update pref. model $\hat{u}$ with $(\boldsymbol{y_1}, \boldsymbol{y_2}, r(\boldsymbol{y_1}, \boldsymbol{y_2}))$
13:    **EXP stage**
14:    $\boldsymbol{x}_{\text{EXP}} \leftarrow \text{argmax}_{\boldsymbol{x}} \text{ qEIUU}$
15:    $\boldsymbol{y}_{\text{EXP}} = \boldsymbol{f}(\boldsymbol{x}_{\text{EXP}})$
16:    Append $D$ with $(\boldsymbol{x}_{\text{EXP}}, \boldsymbol{y}_{\text{EXP}})$
17:    Update outcome model $\hat{\boldsymbol{f}}$ with $(\boldsymbol{x}_{\text{EXP}}, \boldsymbol{y}_{\text{EXP}})$
18:    **GD stage**
19:    $(\boldsymbol{X}_{\text{GD}}, \boldsymbol{Y}_{\text{GD}}) \leftarrow \text{Local Gradient Descent}(\boldsymbol{x}_{\text{EXP}})$
20:    Append $D$ with $(\boldsymbol{X}_{\text{GD}}, \boldsymbol{Y}_{\text{GD}})$
21: **end while**

---

**Algorithm 2** MULTI-GRADIENT DESCENT

---

1: Initialize $\boldsymbol{x}_{\text{GD}} \leftarrow \boldsymbol{x}_{\text{EXP}}$
2: $(\boldsymbol{X}_{\text{GD}}, \boldsymbol{Y}_{\text{GD}}) = (\emptyset, \emptyset)$
3: # of multi-gradient steps, $n_{\text{GD}}$   ▷ `default:10`
4: # of GI optimizations, $n_{\text{GI}}$   ▷ `default:1`
5: Early stopping threshold, $\varepsilon_{\text{GD}}$   ▷ `default:0.1`
6: **for** $i \leq n_{\text{GD}}$ **do**
7:    Compute $\boldsymbol{\mu}^{\nabla}(\boldsymbol{x}_{\text{GD}})$ using (7a)
8:    Compute $M = \boldsymbol{\mu}^{\nabla}(\boldsymbol{x}_{\text{GD}})^{\top} \boldsymbol{\mu}^{\nabla}(\boldsymbol{x}_{\text{GD}})$
9:    $\boldsymbol{\alpha} \leftarrow \text{Frank-Wolfe}(M)$
10:    $\boldsymbol{x}_{\text{GD}} \leftarrow \boldsymbol{x}_{\text{GD}} - \eta \boldsymbol{\alpha}^{\top} \boldsymbol{\mu}^{\nabla}(\boldsymbol{x}_{\text{GD}})$
11:    **if** $\boldsymbol{x}_{\text{GD}} \in \mathbb{X}$ **and** $\left\lVert \boldsymbol{\alpha}^{\top} \boldsymbol{\mu}^{\nabla}(\boldsymbol{x}_{\text{GD}}) \right\rVert_2^2 > \varepsilon_{\text{GD}}$ **then**
12:       Evaluate the true objective: $\boldsymbol{y}_{\text{GD}} = \boldsymbol{f}(\boldsymbol{x}_{\text{GD}})$
13:       Append $(\boldsymbol{X}_{\text{GD}}, \boldsymbol{Y}_{\text{GD}})$ with $(\boldsymbol{x}_{\text{GD}}, \boldsymbol{y}_{\text{GD}})$
14:       Update outcome model $\hat{\boldsymbol{f}}$ with $(\boldsymbol{x}_{\text{GD}}, \boldsymbol{y}_{\text{GD}})$
15:       **for** $j \leq n_{\text{GI}}$ **do**
16:          $\boldsymbol{x}_{\text{GI}} \leftarrow \arg\max_{\boldsymbol{x}'} \text{GI}$
17:          Evaluate the true objective: $\boldsymbol{y}_{\text{GI}} = \boldsymbol{f}(\boldsymbol{x}_{\text{GI}})$
18:          Append $(\boldsymbol{X}_{\text{GD}}, \boldsymbol{Y}_{\text{GD}})$ with $(\boldsymbol{x}_{\text{GI}}, \boldsymbol{y}_{\text{GI}})$
19:          Update outcome model $\hat{\boldsymbol{f}}$ with $(\boldsymbol{x}_{\text{GI}}, \boldsymbol{y}_{\text{GI}})$
20:       **end for**
21:    **else**
22:       **break**
23:    **end if**
24: **end for**
25: **return** $(\boldsymbol{X}_{\text{GD}}, \boldsymbol{Y}_{\text{GD}})$

---

**Algorithm 3** Frank-Wolfe Algorithm

1: **input M**
2: **initialize** $\boldsymbol{\alpha} = [\frac{1}{n_f}, ..., \frac{1}{n_f}]$ s.t. $\mathbf{1}^\top \boldsymbol{\alpha} = 1$
3: **for** $j \leq$ # max no. of Frank-Wolfe steps **do**
4:     $\hat{t} = \arg\min_r \sum_t \alpha_t \mathbf{M}_{rt}$
5:     $\hat{\gamma} = \arg\min_\gamma ((1-\gamma)\boldsymbol{\alpha} + \gamma \mathbf{e}_{\hat{t}})^\top \mathbf{M} ((1-\gamma)\boldsymbol{\alpha} + \gamma \mathbf{e}_{\hat{t}})$
6:     $\boldsymbol{\alpha} = (1-\hat{\gamma})\boldsymbol{\alpha} + \hat{\gamma}\mathbf{e}_{\hat{t}}$
7:     **if** $\hat{\gamma} \sim 0$ **then**
8:         **break**
9:     **end if**
10: **end for**
11: **return** $\boldsymbol{\alpha}$

