# OpenReview forum: "Preference-based Multi-Objective Bayesian Optimization with Gradients"
_NeurIPS.cc/2024/Workshop/BDU — NeurIPS BDU Workshop 2024 Poster_

### Official Review · Reviewer_5wdJ · 2024-09-26
**The review of Reviewer 5wdJ for Submission108**

**Rating:** 7
**Confidence:** 4

**Review:**

## Summary

This paper focuses on the task of Multi-objective Bayesian Optimization (MOBO) and presents Preference-Utility-Balanced MOBO (PUB-MOBO), which aims to refine solutions to be near-Pareto-optimal while remaining efficient and user-oriented. Unlike traditional methods, PUB-MOBO does not require estimating the entire Pareto front and combines multi-objective Bayesian optimization with local multi-gradient descent techniques. Experimental results demonstrate that PUB-MOBO consistently outperforms existing methods in terms of utility regret and proximity to the Pareto front on both synthetic and real-world benchmarks.

## Strengths

This is a good paper for Multi-objective Bayesian optimization (MOBO). Its merits span the following aspects:

1. This paper extend the two stages (preference exploration, and outcome evaluation via experiments) in existing method with an additional stage based on local multi-gradient descent。

2. This paper adapts the GI function in combination with the algorithm flow and gives a rigorous derivation process.

3. This paper proposes a new utility function, Preference-Dominated Utility Function (PDUF), to model user preferences in preference-driven multi-objective Bayesian optimization (MOBO).

4. This paper has strong logic, clear symbol definitions, and strong readability.

## Weaknesses

1. PUB-MOBO combines global and local searches, as well as multi-gradient descent, which may increase the computational complexity of the algorithm. For high-dimensional problems or large-scale datasets, this may result in high computational costs.

2. PUB-MOBO may contain multiple parameters that need to be adjusted. The choice of these parameters has a significant impact on algorithm performance, but the article may not provide sufficient guidance.

---

### Official Review · Reviewer_qTVV · 2024-10-07
**Preference-based Multi-Objective Bayesian Optimization with Gradients**

**Rating:** 8
**Confidence:** 3

**Review:**

This paper presents a novel approach to multi-objective Bayesian optimization (MOBO) called Preference-Utility-Balanced MOBO (PUB-MOBO). Traditional MOBO methods assume all optimal solutions are equally desirable, which isn't always true in practice. PUB-MOBO addresses this by incorporating user preferences into the optimization process. It starts with a global search to identify regions aligned with user preferences, followed by a local search using gradient descent to find dominating solutions near the Pareto-front. The paper introduces a new utility function, Preference-Dominated Utility Function (PDUF), that effectively captures both dominance and user preferences. PUB-MOBO significantly outperforms existing methods in terms of proximity to the Pareto-front and utility regret, demonstrating its ability to identify high-utility, near-Pareto-optimal solutions.

---

### Decision · Program_Chairs · 2024-10-09

Accept (Poster)